# Use of Ultrasonic Cleaning Technology in the Whole Process of Fruit and Vegetable Processing

**DOI:** 10.3390/foods11182874

**Published:** 2022-09-16

**Authors:** Wenhao Zhou, Frederick Sarpong, Cunshan Zhou

**Affiliations:** 1School of Food and Biological Engineering, Jiangsu University, Zhenjiang 212013, China; 2Value Addition Division, CSIR-Oil Palm Research Institute, Kade P.O. Box 74, Ghana

**Keywords:** ultrasonic cleaning, cavitation effect, fruits and vegetables, transducer

## Abstract

In an era of rapid technological development, ultrasound technology is being used in a wide range of industries. The use of ultrasound technology in fruit and vegetable processing to improve production efficiency and product quality has been an important research topic. The cleaning of whole fresh fruits and vegetables is an important part of fruit and vegetable processing. This paper discusses the development process of components of the ultrasonic equipment, the application of ultrasonic technology in fruit and vegetable cleaning, and the research advances in ultrasonic cleaning technology. Moreover, the feasibility of ultrasonication of fruits and vegetables for cleaning from the perspectives of microbial inactivation, commodity storage, and sensory analysis were discussed. Finally, the paper identified the inevitable disadvantages of cavitation noise, erosion, and tissue damage in fruit and vegetable processing and points out the future directions of ultrasonic fruit and vegetable cleaning technology.

## 1. Introduction

Ultrasound became a new concept in 1922 when the definition of ultrasound was first introduced. Ultrasound technology has developed rapidly since the 1950s and is receiving increasingly close attention. Ultrasound applications can be divided into two main categories: low-power ultrasound for ultrasonic testing and diagnosis and high-power ultrasound for the performance and state of materials, such as ultrasonic cleaning, welding, processing, grinding, and atomization [1,2]. Ultrasonic cleaning has become the most widely used high-power ultrasonic technology. It has gradually expanded from its initial tight application in machinery and electronics to emerging areas such as the medicine, food, chemical, aerospace, and nuclear industries [3,4,5]. The principle of ultrasonic cleaning lies in the ultrasonic waves generated in water to produce enough “cavitation bubbles” and the release of energy to complete the cleaning operation. Ultrasonic cleaning is also known as “brushless cleaning” due to its unique cleaning method [5].

Fruits and vegetables are rich in vitamins, dietary fiber, and essential minerals. In recent years, the consumption of fruits and vegetables has been increasing rapidly due to the pursuit of healthy living and the upgrading of consumer attitudes [6]. The washing process is a key part of fruit and vegetable processing, and it is closely related to the quality of the fruits and vegetables [7]. Traditional fruit and vegetable cleaning equipment such as brushes, drums, water streams, and air baths can remove dirt from the surface of fruits and vegetables through friction, bubbles, water flow, and material coupling. Widely-used cleaning equipment are brushes and rollers because of their powerful cleaning power as well as their excellent cleaning efficiency, but this type of cleaning equipment is not suitable for cleaning leafy fruits and vegetables—where the tissue surface can easily be severely damaged during the cleaning process. On the other hand, air bath and immersion nozzle cleaning methods are less efficient and do not guarantee product quality. In recent years, ultrasound has been increasingly used in the food processing field, but its research in fruit and vegetable cleaning is still in the exploration stage [8]. The objective of this paper is to present the latest developments in the application of ultrasonic cleaning of fruits and vegetables. The review contents include the following two aspects: (1) an overview of the trends in ultrasonic technology in fruit and vegetable cleaning from the point of view of mechanical construction and process implementation, and (2) the shortcomings of ultrasonic cleaning technology in fruit and vegetable processing, and its future developmental trend were pointed out.

## 2. Ultrasonic Cleaning Technology and Application

### 2.1. Ultrasonication

Sound waves are the form in which the mechanical vibration state (or energy) of an object propagates in a conducting medium. These sound waves are divided into three categories in the sound spectrum: infrasound (v < 16 Hz), acoustic waves (16 Hz < v < 16 kHz), and ultrasound (v > 16 kHz). Ultrasound is a mechanical wave with an extremely short wavelength—generally shorter than 2 cm in air. The ultrasonic frequency usually uses ranges above 20 kHz, i.e., the particles vibrate more than 20,000 times per second, which exceeds the upper limit of the frequency of ordinary human hearing (16 kHz) [9]. Power, intensity, and energy density are important parameters of ultrasound. Ultrasound techniques are frequently used in different areas, such as the determination of texture, viscosity, solid and liquid content; degassing of liquid foodstuffs; dispersion of polymeric materials; disruption of cells; and production of emulsions as well as induction of enzyme deactivation [10].

### 2.2. Principles of Ultrasonic Cleaning Technology

Ultrasonic cleaning was initially applied to surface cleaning in the electronics industry, and it has been now gradually applied to food manufacturing [11]. The principle is based on the “physical effect” created by ultrasound to clean surface dirt. At the same time, the hydrogen peroxide active compound produced by ultrasound inactivates microorganisms. In recent years, ultrasound technology has been widely used for food processing and manufacturing operations. Ultrasound inactivates microorganisms by promoting the growth of cavitation bubbles in liquid media and propagating them through the cellular structure [12]. 

The principle of ultrasonic cleaning is to produce a cavitation effect through the liquid medium to remove the dirt particles. When the ultrasonic wave propagates in the liquid medium, it produces mechanical vibrations and acoustic flow, thereby converting acoustic energy into mechanical energy. The gas dissolved in the liquid medium expands under the action of ultrasonic waves, completing a series of reactions—such as the formation and rupture of cavitation bubbles. At the same time, cavitation caused by changes in temperature and pressure will also cause chemical changes that accelerate the cleaning efficiency [13]. When turbulent microcirculation is generated near a solid surface, a microjet with a velocity of approximately 110 m/s and a high impact force is produced, resulting in a collision density of up to 1.5 kg/cm^2^ [14]. Figure 1 shows the ultrasonic cleaning process for cavitation bubbles.

### 2.3. Ultrasound Fields and Cavitation Effects

Ultrasonic cavitation has certain drawbacks; for example, standing waves are always present in the ultrasound field, which can lead to an uneven distribution of the sound wave and seriously affect the ultrasonic cavitation effect. In order to improve the effectiveness of ultrasonic cavitation and promote industrial applications, many scholars have started to study acoustic–chemical reactors. Dion used high-powered aggregated ultrasound to increase the absolute acoustic pressure of the acoustic–chemical reactor, thus expanding the area of acoustic cavitation [15]; Cintas bonded multiple high-throughput ultrasonic transducers to the inner wall of the reactor to improve the effectiveness of ultrasonic cavitation [16]; and Asakura developed a large acoustic–chemical reactor with an ultrasonic frequency of 500 kHz and a power of 620 W. The physics were mapped, and the reaction system was evaluated using the dose-thermal method. The results showed that although the absolute acoustic pressure in the reactor was high and the acoustic–chemical yield was relatively high, the electro-acoustic conversion efficiency was 70% and the acoustic energy consumption was heavily dispersed [17]. 

However, little attention has been paid to the physics distribution characteristics within the acoustic–chemical reactor in the above studies. It is well known that the ultrasonic cleaning process is essentially a multi-physics coupled process, where the interacting structural, ultrasonic, and fluid fields produce different cavitation effects in different ultrasonic modes, which have an impact on cleaning efficiency and microbial activity. In recent years, to better explain the mechanism of action of ultrasonic cavitation, the characteristics of the physical distribution in acoustic–chemical reactors have become a hot issue for researchers, but fewer studies have been reported on the numerical simulation of multi-physics field coupling in the ultrasonic cleaning process. Zhang studied the physical distribution characteristics of the composite sound source in different directions using the aluminum foil etching method and used the sound pressure intensity and sound field uniformity as important indicators for evaluating the ultrasonic cavitation effect [18]. To investigate the effect of ultrasound power on the cavitation effect at the characteristic sampling points, potassium iodide (KI) imaging and dichloromethane degradation methods were used to quantitatively map the cavitation field [19]. Hallez evaluated a variety of methods, such as hydrophone and laser chromatography, to characterize the distribution of cavitation fields in the HIFU acoustic–chemical reactor [20]. When mapping the sound field using the hydrophone, it was found that ultrasound waves were constantly reflected and superimposed in the reaction chamber, creating a standing wave effect that resulted in a very uneven distribution of the sound field and the formation of a “cavitation blind zone” [21]. In addition, the shape of the ultrasound probe significantly impacted the physical field distribution characteristics and cavitation effects. 

To characterize the acoustic intensity distribution of an ultrasound field, traditional test methods qualitatively quantified the physical field by measuring the chemical yield of acoustic cavitation; however, this method can only reflect the cumulative cavitation effect over a period of time and cannot describe changes in the physical field in real time. In addition, traditional measurement methods have the disadvantages of low spatial resolution, poor repeatability, rigorous measurement processes, and high measurement costs. To achieve real-time, non-destructive, and high-resolution measurement, Wang used an optical interferometer to accurately obtain real-time distribution characteristics of the sound field by detecting the velocities of plane and spherical wave particles in the sound field through thin films [22]; C. Koch applied a fibre-optic probe sensor to measure the sound field in an ultrasonic cleaning tank in real-time and detected changes in the optical path of the fibre-optic probe by means of an external differential interferometer to obtain a sound velocity-dependent signal. The results showed that the sound velocity signal is similar to the cavitation noise signal and the piezoelectric signal, and that the fibre-optic probe sensor can replace the hydrophone for accurate measurement of the high spatial resolution sound field [23]. D. Fan used an acoustic intensity meter (AIM) to study the effect of ultrasound intensity on surimi glue and the distribution characteristics of the sound field in an acoustic–chemical reactor. The results showed that the different ultrasonic parameters, fluid medium, and vessel material can change the sound field intensity, and the ultrasonic frequency has the most obvious effect on the sound intensity. Specifically, the ultrasonic intensity increases with the height of the fluid medium, and the volume of the medium increases before decreasing; the ultrasonic intensity is highly non-linear with the ultrasonic frequency, and the secondary structure of the muscle fibres changes when the sound intensity of the sound field reaches a certain level, resulting in an increase in the gel strength of the surimi [24]. 

With the development of high-performance computing, researchers have started to use numerical simulation techniques to study the physical field distribution characteristics in acoustic–chemical reactors. Zhai evaluated three orthogonal ultrasonic waves to numerically simulate the internal sound field of a liquid medium [25]. Laish performed numerical simulations of the flow field of the ultrasonic cleaning process and studied the decontamination process in combination with laser Doppler velocimetry (LDA) tests [26]. Servant performed numerical simulations of ultrasound-induced bubble dynamics and graphically depicted the evolution of cavitation bubbles in the time domain [27]. Lebon studied the transient pressure field in cavitating media and found that numerical models could better characterize cavitation in the near field, while the far field region was difficult to capture accurately due to the “cavitation shielding” effect [28]. 

To find the exact resonant modes of physics within a non-linear acoustic framework, Louisnard discussed the effect of different boundary conditions on the resonant frequencies of the system, based on numerical simulation techniques [29]. Klíma used finite element (FEM) methods to verify the existence of standing wave effects and to optimize the mechanical design of an acoustic–chemical reactor with an ultrasonic frequency of 20 kHz. He argued that constructing a multi-physics model within an acoustic–chemical reactors require accurate modelling of the excited sound source and the propagating medium, which means that the relevant physical fields must be coupled [30]. Vanhille used numerical simulations to develop a multi-physics field coupled model of a cavitation bubble swarm in a non-linear standing wave field to study the forces on the cavitation bubble swarm in a non-linear coupled system. The results showed that the cavitation bubble cluster is mainly subjected to a first-order Bjerknes force caused by the pressure gradient of the oscillating bubble. When the acoustic pressure amplitude was small, the first-order Bjerknes force depended on the resonant frequency of the system; as the acoustic pressure amplitude increased, the first-order Bjerknes force had a significant effect on both bubble motion and aggregation patterns [31]. Zhang Zhiqiang applied the fluid–solid coupling (FSI) method to establish a multi-physical field coupling model for an acoustic–chemical reactor and systematically investigated the effect of sound field orientation on the sound field distribution characteristics. The results showed that optimizing the sound field orientation can significantly improve the absolute sound pressure and sound field uniformity compared to simply increasing the number of transducers. Compared with a unilateral radiation array, the opposite radiation of the transducer can effectively mitigate the standing wave effect, increase the sound intensity in the sound field by a factor of 30, cavitate an area of more than 95% of the overall space, and improve the extremely uneven sound field distribution [18]. Wei used numerical simulations to establish a coupled multi-physical field model of the cavitation process and investigated the effects of different ultrasound modes, transducer arrangements, and probe shapes on the acoustic–chemical physical field distribution characteristics. Combining simulation and experimental validation, a new ultrasonic probe with multiple stepped axes was optimally fabricated to produce multiple cavitation zones for a more uniform acoustic field distribution [32]. Sajjadi investigated the effect of ultrasonic power on the volume fraction and fluid flow pattern of microbubbles and analyzed the formation process and kinetic characteristics of cavitation bubbles using mathematical models, CFD simulations, and particle image velocimetry (PIV) tests. The mathematical model analysis showed that ultrasonic power significantly affected the formation conditions and oscillation velocity of cavitation bubbles. CFD simulations showed that the total volume of cavitation bubbles increased by approximately 4.95% for every 100 W increase in ultrasonic power [33]. Therefore, we can consider that computer-aided software provides an effective method for the in-depth study of ultrasonic cavitation effects and multi-physical field coupling models.

### 2.4. The Model of Ultrasonic Cleaning Technology

#### 2.4.1. The Cavitations Model

Ultrasonic pressure causes the bubble to contract, thereby increasing the temperature, on the assumption that the hydrostatic pressure *P*_1_ in the air is equivalent to that in the bubble and is compressed by the adiabatic process [34]. Before the bubbles contract, the pressure was *P*_0_ and the water temperature was *T*_1_. The ratio of pressure heat to volume heat was *δ*, where *δ* was taken as 1.33. The internal temperature *T*_0_ was calculated using Equation (1):(1)$$T_{0}=T_{1}\left(\delta -1\right){\left(P_{0}/P_{1}\right)}^{\left(-1/\delta \right)}$$

#### 2.4.2. The Ultrasonic Output Model

*P* is a symbol for sound intensity:(2)P=12ρCωα2g/sec3

*C* is the number of ultrasonic longitudinal waves, ultrasonic angular velocity ω = 2πf, ρ is the density of the auxiliary medium, and a is the amplitude of the oscillator.

The following is the expression of the relationship between radiative sound pressure *P* and *M*:(3)$$M=1/2\left(1+\sigma \right)\left(P/C\right)\left(kg/cm\cdot sec^{2}\right)$$

#### 2.4.3. The Efficiency Model

The cleaning rate *CL* is measured in terms of efficiency, as follows:(4)$$CL\%=\left(CL_{f}/CL_{i}\right)\times 100\%$$
where *CL_f_* is the number of clean vegetables and *CL_i_* is the number of vegetables.

*D* is the damage rate:(5)$$D\%=\left(D_{f}/CL_{i}\right)\times 100\%$$

*D_f_* is the number of damaged vegetables.

### 2.5. The Main Equipment for Ultrasonic Cleaning

The ultrasonic cleaning equipment is equipped with cleaning tanks, rinsing tanks, mechanical drain valves, and jet valves, while the exterior is equipped with an automatic washing liquid device, storage tank, washing pump, and an infusion tube located at the bottom. The storage tank is connected to the interior of the ultrasonic cleaning tank through the infusion tube. The upper part of the rinsing and washing tanks are each equipped with automatic spray sensors. The frame structure has a trolley that matches the rails and a load cell. The ultrasonic vegetable cleaning equipment is composed of two parts: an ultrasonic generator and an ultrasonic vibration box (Figure 2).

The transducer is the main component of the ultrasonic cleaning device, and the research on the transducer in the field of ultrasonic applications has a relatively mature theory. The devices can convert electrical energy into sound energy. In 1917, Langevin invented the sandwich piezoelectric transducer, and in 1933 the magnetoelectric transducer replaced the piezoelectric transducer with high strength and stability. In 1950, Jaffe used lead zirconate titanate (PTZ) as a piezoelectric material and found that it has higher mechanical strength and electromechanical conversion rates. In 1956, Mason invented a connection between ultrasonic amplitude rods and piezoelectric transducers to obtain more significant vibrational displacement and amplitude, which played a role in efficient fusion energy transfer [35]. In the ultrasonic cleaning process, the performance of the transducer directly affected the cleaning effect. The ultrasonic energy during ultrasonic cleaning is transmitted through a transducer-driven wash tank base plate. Therefore, the vibration characteristics have a very important impact on the effect of ultrasonic cleaning and have attracted several studies in this regard. Also, the sound field generated by the transducer has a direct relation with the cavitation effect of the ultrasonic waves. Silva proposed a switching-free multi-band impedance matching network technique based on multi-resonant circuits, which was observed to provide short-circuit and open-circuit conditions at specified frequencies, thereby allowing capacitors and inductors to form multi-band impedance matching networks [36]. Currently, piezoelectric ultrasonic transducers are widely used and the requirements for ultrasonic measurement accuracy, measurement range, ultrasonic power, and miniature devices are increasing. The limitations in the development of ultrasonic cleaning technology are the lack of suitable, reliable, economical, and durable ultrasonic transducers, therefore the research of ultrasonic cleaning-related equipment is also very critical [5].

The above research background shows that researchers have found that the physics distribution in the acoustic–chemical reactors directly affect the cavitation characteristics and cleaning effect through experimental observation and numerical simulation. Based on the above situation, by establishing the evaluation index of the sound field distribution characteristics in the cleaning process, the multi-field coupling model focuses on the power degree, ultrasonic frequency, dynamic viscosity, system flow, transducer spacing, and the influence of its array arrangement on the sound field distribution characteristics, and further clarifies the mechanism of ultrasonic waves. It also provides theoretical guidance and technical support for visual analysis and quantitative evaluation of the ultrasonic cleaning process, scientific design of ultrasonic equipment, and promotion of the industrial application of power ultrasound.

## 3. Effect of Ultrasonic Cleaning on Microorganisms on the Surface of Fruits and Vegetables

Contamination can occur throughout the production and sale of fruits and vegetables. Microorganisms are one of the primary pollutants, causing higher requirements for food safety [37,38]. Cleaned fruits and vegetables will not only reduce surface microorganisms but are also expected to have a longer shelf-life. 

The reduction of microorganisms on the surface of fruits and vegetables is the result of a combination of two causes: (1) microbial shedding, where microbial clumps or contaminants (e.g., clay) are removed from the surface of fruits and vegetables by acoustic field forces after sonication; and (2) microbial inactivation, where microbial populations are reduced by inactivation through a series of physical, mechanical, and chemical effects produced by ultrasonic cavitation [39,40].

In studies of microbial inactivation, ultrasonic treatment could produce sound cavitation in the cells and induce the thinning of the cell membrane to inhibit the reproduction of microorganisms [41]. Similarly, heating could produce cavitation and hydroxyl radicals [42]. The spaces between the liquid molecules create bubbles in a continuous cavitation cycle. These bubbles are divided into two categories according to their structure: the first type of bubbles is non-linear, similar to atmospheric clouds, and the persistent bubbles that form under the action of pressure cycles. The second category of bubbles with internal (transient) cavitation are unstable, smaller, and of very short duration [43,44]. The process of cavitation bubbles growing and bursting increases the pressure and temperature in the medium, creating a shockwave. The ability to release is enormous, generating enormous shear forces and destroying the cell walls and membrane structures to kill microbial cells [42].

The second cause of microbial death is the chemical reaction caused by ultrasonic treatment. In fact, due to the antibacterial mechanism of hydroxyl radicals, ultrasonic decomposition using a 20 kHz ultrasound unit can enhance the inactivation of microorganisms [45]. Researchers showed that ultrasound action raised the temperature inside the bubble, which in turn accelerated the production of hydroxyl radicals. In other words, the leading causes of ultrasonic cavitation in chemical sterilization are free radicals, single-electron transfer, and recombination of hydroxyl radicals and hydrogen atoms, thereby effectively reducing the number of bacteria [46]. Chemical reagents can be used to assist sterilization technology, thus, by adding other chemicals to the liquid medium, a series of chain reactions will increase the sterilization effect of cavitation bubbles. In addition, hydroxyl radicals (hydroxides) can react with the sugar–phosphate backbone of the DNA strand, leading to the separation of phosphate–ester bonds and the breakage of double-stranded microbial DNA [47].

The residual microorganisms on the surface of fruits and vegetables after ultrasonic cleaning are shown in Table 1.

## 4. Analysis of Fruits and Vegetables Cleaning Process

### 4.1. The Influence of Various Factors on Ultrasonic Cleaning

Cleaning equipment, cleaning agents, cleaning processes, and the nature of the fruits and vegetables are the four main factors affecting ultrasonic cleaning, while other possible factors can also have an impact on cleaning efficiency. Figure 3 shows the factors affecting ultrasonic cleaning.

The effectiveness of the cavitation effect is closely related to ultrasonic cleaning, and many scholars have carried out research on the factors influencing the cavitation effect from the perspective of cavitation bubble dynamics. They also explored the cleaning effect from the acoustic parameters of cleaning equipment. For instance, Worapol and Jatuporn conducted a harmonic response analysis of the cleaning tank using different placement positions of the transducers at 28 kHz and 40 kHz. It was discovered that the transducers could obtain a higher sound pressure intensity than when the whole bottom surface and transducer were placed simultaneously on the ground [81].

In addition, Niemczewski studied the effect of the pH of the cleaning medium on the cleaning effect and concluded that the addition of sodium sulphite makes the medium less alkaline and more effective in cavitation [82]. For further exploration, Niemczewski and Kołodziejczyk analyzed the cavitation effect with an acidic solution between 1% and 7% (*w*/*w*) and found that a gradual increase in the acidic concentration significantly increased the intensity of the cavitation effect [83]. Kim directly explored the collapse mechanism of cavitation bubble rupture from the microscopic level and found four oscillation behaviours, including volume, shape splitting, and the chaotic oscillation behaviour of cavitation bubble ruptures [84].

Meanwhile, the ultrasonic cleaning parameters may also be decided by the natural characteristics of fruits and vegetables, such as various kinds of microorganisms from them. Therefore, a detailed evaluation of all treatment parameters is necessary in order to obtain a higher cleaning efficiency. For the effective killing of bacteria, Takundwa found that washing lettuce with a combination of 771.2 IU/g of nisin and 0.185% *v*/*v* oregano for 14.65 min of ultrasound resulted in maximum removal of *E. coli O157:H7* and *Listeria monocytogenes* [85]. However, Zhang found that after treating blueberries with ultrasound alone for 10 min, the blueberry surface was reduced by approximately 2 log CFU/g of sterile *Listeria monocytogenes* and 3–4 log CFU/mL of viable bacteria remained in the wash water. Afterwards, he added 2 mM of carvacrol for auxiliary washing and the bacteria count on the blueberry surface was significantly reduced [86]. Therefore, when dealing with different types of fruits and vegetables, it is important to set the appropriate ultrasonic cleaning parameters according to their nature.

### 4.2. Ultrasonic-Assisted Cleaning Technology

The low antimicrobial efficiency of ultrasonic cleaning alone in most cases has been mentioned in several published papers. In contrast, the use of ultrasound in combination with auxiliary reagents may be an effective method for enhancing microbial inactivation and has been previously demonstrated in experiments. 

#### 4.2.1. In Combination with ClO_2_

As a commonly used disinfectant, ClO_2_ has the following advantages: strong bactericidal ability, fast disinfection, strong oxidation resistance, few by-products, wide application pH range, and bactericidal performance remains unchanged in a wide range (pH 3–10) [87]. ClO_2_ is highly soluble in water and can kill many pathogenic bacteria, such as *Escherichia coli* and *Staphylococcus aureus* at different concentrations (0.1 ppm). It kills almost all microorganisms, such as bacterial propagules, hepatitis viruses, phages, and bacterial spores, even when disturbed by organic matter [88]. Chlorine dioxide has strong oxidation, so it has the functions of sterilization, bleaching, deodorization, disinfection, and preservation. The United States and China have approved ClO_2_ as a disinfectant for fruits and vegetables [77]. According to the hygienic standard for chlorine dioxide disinfectant (GB 26366-2010), the allowable concentration of chlorine dioxide for fruits and vegetables cleaning is 100–150 ppm, and the recommended cleaning time is between 10–20min.

The combination of chlorine dioxide and ultrasound can effectively improve the cleaning effect of ultrasound. For example, studies have shown that when chlorine dioxide and ultrasound were combined to clean apples, bacteria was effectively removed from the surfaces [89]. Milan-Sango also tested the same combination of washing methods for the inhibition of *E. coli* and *Salmonella,* and the results showed a significant reduction in the microorganisms associated with bean sprouts [77]. Similarly, Zhao et al. used this technique to treat plum fruits for 10 min, and the experimental results showed that it effectively extended the shelf-life of fruits and reduced the loss rate of vitamin C and flavonoids [11].

In conclusion, ultrasonication combined with ClO_2_ technology effectively inactivated bacteria and viruses. However, chlorine dioxide also has disadvantages, such as high costs, instability, complex monitoring methods, pH sensitivity, and toxicity of disinfection by-product chlorite.

#### 4.2.2. In Combination with Ozone

Ozone has strong oxidation and bactericidal properties and is ranked second in the redox potential in water after fluorine. Ozone is widely used in disinfection, decolourization, deodorization, and oxidative decomposition. Ozone and anions have excellent preservation properties, so the use of ozone technology can significantly extend fruit and vegetables’ freshness and storage time. The low concentration of ozonated water is highly efficient in sterilization and free from secondary pollution. At the same time, ozone can effectively oxidize pesticides on the surface of fruits and vegetables, thereby reducing pesticide residues [90]. The combination of ozone and ultrasonic treatment increases the dissolution rate of ozone in the liquid medium. The latter can directly destroy microorganisms’ RNA and DNA material and preserve the food organoleptic properties [91]. In addition, when tomatoes were washed with an ozone concentration of 0.4 ppm, the organoleptic properties of the tomatoes were intact, and the microorganisms are inactivated at the same time [92]. In experiments with strawberry cleaning, when strawberries were cleaned with ozone and ultrasound, and ozone and chlorine dioxide for 5 min, the results showed that only the ozone treatment resulted in the decolorization of the strawberries, but the combination of ultrasound and ozone treatment had a better cleaning effect [93]. Similarly, when cleaning lettuce with ultrasonic waves and ozone—compared with ultrasonic waves alone—pesticide residues on the leaf surface of the lettuce were considerably reduced, and the quality of the lettuce was not significantly affected [94].

So far, ozone combined with ultrasound technology has been suitable for the application of water in fruit and vegetable washing. Ozone, however, is a very active, unstable, and colourless gas that is highly-oxidizing and will revert to the properties of stable oxygen when in contact with other substances. In order to not damage the sensory quality of food, thresholds for ozone concentration and exposure time should be strictly controlled [95]. In addition, different types of fruits and vegetables have a time threshold for ozone treatment that requires strict control of their exposure time, and the Occupational Safety and Health Administration (OSHA) of the United States requires workers to work no more than 8 h a day continuously at an ozone concentration of 0.1 ppm [92].

#### 4.2.3. In Combination with Electrolysis of Water

Electrolyzed water is also known as electrolytic ionized water or REDOX potential water. Under certain conditions, the acidic electrolytic water produced by electrolysis can be used for disinfection. According to the principles of electrolysis, at lower pH values (pH < 2.7), oxygen is produced at the electrodes and will combine with the chloride to produce an aqueous solution of hypochlorite or chlorite ions. Electrolytic water is acidic electrolytic water (AEW) with oxidizing capacity produced at the anode in a diaphragm electrolytic cell with a certain concentration of electrolyte. The pH of AEW is 2–3, and the ORP of REDOX potential is more than 1100 mV. Reductive alkaline electrolytic water (BEW) is generated at the cathode. The pH of BEW is 10–12, and the ORP of REDOX potential is less than 700 mV. Weak acid electrolytic water (SAEW) is an acidic aqueous solution with hypochlorous acid as the main bactericidal component. It is produced by adding low concentrations of hydrochloric acid and or sodium chloride to soften water and electrolyzing it in a diaphragm electrolytic cell. The pH of SAEW ranged from 5.0–6.5, and the ORP of REDOX potential was more significant than 800–900 mV [96,97,98,99]. Figure 4 shows a schematic diagram of systems producing AEW, BEW, and SAEW.

Electrolytic water reacts with organic matter to generate ordinary water, effectively reducing environmental pollution. The electrolysis of water and HClO can destroy the cell membrane and respiratory transport chain of microorganisms. Electrolytic water combined with ultrasonic treatment is conducive to accelerating cell bacteria death and minimizing early damage to cell bacteria and can also be considered a new, more environmentally-friendly sterilization technology [100]. The AEW method can effectively kill microorganisms after cleaning broccoli for 20 s, and the washing effect was better when electrolytic water was used with an ultrasonic wave [101]. Li et al. found that the activity of *Staphylococcus aureus* decreased significantly after the combined treatment of electrolysis water and ultrasound, and it was significantly better than SAEW alone [101]. When processing cherry tomatoes and strawberries, the total number of aerobic bacteria decreased by 1.77 and 1.29 log, respectively, and the number of yeasts and moulds decreased by 1.50 and 1.29 log, respectively [102]. In other studies, the reduction of *Bacillus cereus* was 3.0 log CFU/g when confectionery was decontaminated with ultrasound and SAEW [103]. Combining ultrasonic and electrolyzed water to clean fruits and vegetables can effectively reduce the number of microorganisms and improve the efficiency of ultrasonic cleaning.

#### 4.2.4. In Combination with Non-Thermal Plasma

Non-thermal plasma (NTP) is a disinfectant produced by dielectric barrier discharge, plasma radio, and corona discharge [104]. Charged particles that bombarded microorganisms in NTP can directly destroy their proteins, nucleic acids, and other macromolecules, resulting in the death of the microorganisms [105]. Active substances in NTP (such as reactive oxygen species, free groups, etc.) react with proteins and nucleic acids in microorganisms, disrupting the normal function of the microorganisms [106,107].

Experiments showed that cold plasma treatment of apples effectively reduced the number of *E. coli* and *salmonella* residues [108]. NTP has also been used in combination with ultrasound in other studies. For example, Liao investigated the effects of different cleaning methods on *Staphylococcus aureus* using ultrasound treatment alone, NTP treatment, and ultrasound and NTP treatment in combination. They showed that ultrasound combined with NTP treatment could accelerate cell death and minimize early damage bacteria [109].

The characteristics of the above four disinfectants that can be used for ultrasonic cleaning are shown in Table 2.

## 5. The Effect of Ultrasonic Cleaning on Sensory and Storage

### 5.1. Sensory Aspects

It is well known that the texture, flavour, colour, and other characteristics of washed fruits and vegetables are the main aspects affecting sensory evaluation. In terms of texture, ultrasonication can change the texture by altering the microstructure of the fruit and vegetables; in terms of flavour, the free radicals generated by ultrasonic cavitation are expected to reduce the off-flavours of the food; in terms of colour, ultrasonication has no strong oxidizing properties compared to disinfectants and has no effect on the colour of the fruits and vegetables [46]. Alexandre found that strawberries treated with ultrasound (35kHz) retained 16% more firmness than samples washed with water [50]. A restricted number of studies have found that ultrasound can catalyze the degradation of off-flavours in wine [46]. Alexandre discovered that the colour of the strawberries was maintained by treating them with ultrasound (35 kHz) for 2 min compared to other treatment methods [50].

In order to obtain a better organoleptic evaluation, it is essential that the ultrasound treatment parameters are set appropriately. For example, the time of sonication is one of the key factors. Rosário et al. noticed that there were no significant differences in colour when strawberries were washed with ultrasound (40 kHz, 500 W) in combination with 40 mg/l of peroxyacetic acid for 5 min. [110]. In contrast, Gómez-López et al. found that orange juice produced significant changes after 6, 8, and 10 min of application of ultrasound (20 kHz, 500 W) [111].

### 5.2. Storage

Fruits and vegetables are difficult to store for long periods of time because they become highly perishable when subjected to water loss, microbial infection, or mechanical damage [112,113]. The ultrasonic treatment can be assisted to prolong the shelf-life of fruits and vegetables provided, but not cause damage to them. The main principle of fruit and vegetable storage is the inhibition of key enzyme activities of polyphenol oxidase and peroxidase after ultrasound treatment. Ultrasound treatment effectively maintains the integrity and permeability of cell membranes, while improving antioxidant capacity during storage. Zhang et al. revealed that the use of 0.5% zinc acetate in combination with ultrasonic cleaning extended the shelf-life from 2 to 8 days at 4 °C. The cleaning technique did not only delay the proliferation of microorganisms but was also beneficial in maintaining the quality of fresh-cut cauliflower during storage [76]. Zhang et al. found that both polyphenol oxidase and peroxidase activities of bok choy were effectively inactivated by the synergistic effect of ultrasonic (30 kHz) treatment for 10 min (UT-10) and aeration packaging (MAP) during 30 days of storage at 4 °C. The polyphenol oxidase activity of the samples treated with UT-10 + MAP was lowest at 15 days storage. These results suggest that the use of sonication prior to MAP is effective in maintaining the quality of bok choy during storage [114]. Although studies have shown that the shelf-life of fruits and vegetables is expected to be extended after ultrasound treatment, the preservation effect varies for different ultrasonic powers. Further research is necessary to optimize the application of ultrasonic cleaning technology in extending the shelf-life of fruits and vegetables.

## 6. Consumer Acceptance

In general, consumers are keen on products that are good-looking, hygienically safe, nutritious, and have little variation in organoleptic quality [115]. Ultrasound is generally considered to be safe and harmless and has clear advantages over simple rinsing or chemical cleaning, thereby offering the potential for sustainability. Numerous studies of potential applications in the food industry suggest that ultrasound technologies and products should be evaluated and accepted by consumers. Even though many factors may influence consumer acceptance of food innovations, there will certainly be a gradual acceptance of ultrasound technology over time, for example when electric lamps replace oil lamps. This is the inevitable trend of history.

## 7. Shortcomings and Perspectives of Ultrasound Technology

Although ultrasonic cleaning technology for fruit and vegetable processing has many advantages, there are still some disadvantages that need to be improved, such as long-term ultrasonic cleaning time, which causes cavitation erosion, and high-frequency ultrasonic cleaning, which damages the outer layer of fruits and vegetables tissue and causes nutrient loss. Moreover, the long-time cavitation noise may affect the human health of workers. 

### 7.1. Cavitation Erosion

During ultrasound treatment, high- and low-frequency ultrasound is often used alternately to reduce the effect of standing waves. Each frequency of the ultrasonic generator has an adjustable range. Still, when the maximum power input exceeds the bearable range of fruits and vegetables, the surfaces of the fruits and vegetables are vulnerable to damage. In addition, ultrasonic cleaning equipment can also cause surface damage to mechanical equipment due to long-term operation. Therefore, exploring the frequency limits of cleaning different fruits and vegetables is crucial to optimize the ultrasonic cleaning process [116]. 

The cause of erosion is mainly the microjets produced by bubble cavitation destroying the surface by eroding surface particles, while the cavitation microjets may be made by plastic deformation or material fracture [117,118]. It has been confirmed that bubble growth and collapse cycles during cavitation can generate shear stresses, and microjets can produce thin, high-speed shear layers. Based on the previous evidence, we can assume that tensile, compressive, and shear stresses can be generated at high pressure [119]. 

### 7.2. Tissue Damage

In the ultrasonic cleaning process of fruits and vegetables, the rupture of cavitation bubbles leads to local high temperatures and high pressures of fruits and vegetables. Instantaneous hyperthermia leads to the rupture of tissue cell membranes, causing tissue softening. At the same time, tissue damage can also lead to changes in enzymatic reactions, thereby reducing the shelf-life of fruits and vegetables and seriously affecting the quality. The difference in the quality of fruits and vegetables caused by ultrasound is related to the rupture of the cavitation bubbles, where the effect of tissue damage caused by low-frequency and high-intensity ultrasound is obvious. The lower the frequency and the higher the power, the greater the degree of damage caused by ultrasound to biological tissue [120,121]. Muzaffar showed that after sonicating cherries for more than 20 min, the hardness and vitamin C levels decreased by 3% and 5%, respectively, which are lower than in the control samples [122]. Zhu showed that after washing cabbage leaves with ultrasound, there was significant tissue damage on the leaves and a reduction in the amount of vitamin C. In addition, when the ultrasound frequency was adjusted to 135 kHz, leaf tissue damage improved significantly, indicating that changing the appropriate ultrasound parameters can reduce tissue damage in fruits and vegetables [123]. 

### 7.3. Cavitation Noise

“Cavitation noise” refers to a series of reactions that lead to bubble growth, oscillation, and rupture when ultrasonic waves propagate through a medium. When the cavitation bubble bursts, the energy stored inside is released, and noise is generated [124]. Workers who have worked in ultrasound for a long time may develop a disturbance of the auditory nerve and characterize a range of symptoms, including headache, nausea, vomiting, fatigue, and temporary tinnitus, known as “ultrasound disease.” Iversen believes that ultrasound-induced hearing loss is due to the activation of the vestibular otolithic organs by acoustic radiation forces [125]. Ryo Takagi et al. tested a novel filtering method that differs from traditional frequency-domain filters by selectively eliminating the high-intensity noise component of RF imaging signals. When applied to the cavitation-enhanced focusing of cavitation sound signals from cavitation bubbles in the high-intensity focusing region, this method reduces cavitation noise without affecting the cavitation effect of the liquid medium [126]. At this stage, there is no data to confirm the specific damage of cavitation noise to the human body, but this does not mean that we can ignore the impact of cavitation noise on human health. Further exploration is needed in this direction.

### 7.4. Commercial Prospects

Many techniques have been used for the microbial decontamination of food surfaces, but currently more chemical reagents are used, which has some problems. Physical methods are more likely to win the hearts of consumers than chemical methods in the future. While ultrasound technology or chemical treatments have been successfully used independently in a variety of food processing applications, their combination has not been widely tried and the future promises to reduce the use of chemical reagents and improve the quality and safety of fruit and vegetables [127].

In terms of mechanical considerations, the flow unit design, number of transducers, piping arrangement, etc., are unique for each ultrasound device and can be patented to effectively safeguard commercial intellectual property [128].

### 7.5. Development Trends

Ultrasound technology has been proven to have an important role in fruit and vegetable decontamination and promises to be a green and novel technology to replace conventional washing with chlorine solutions. Although low-frequency ultrasound has been widely used in fruit and vegetable washing, high-frequency ultrasound washing has not yet been covered, which also holds promise for future research. In addition, reasonable treatment parameters should be set according to the type of fruits and vegetables and more research into the commercial application of ultrasound should be carried out.

## 8. Conclusions

Although ultrasonic cleaning has many advantages, it is not devoid of defects, such as noise during cleaning. The problem of cavitation noise is not sufficiently pronounced in small machines but is particularly prominent in large industrial production. Therefore, eliminating or reducing cavitation noise is the focus of future research work. For cavitation phenomena, we can study the specific cavitation conditions of different materials and discover the optimization environments needed to reduce surface roughness. Ultrasound may cause undetectable damage to the interior of fruits and vegetables and it is recommended that these are identified by ultrasonic flaw detection techniques.

Ultrasonic frequency, sound power density, temperature, cleaning media, placement, and other basic factors greatly impact ultrasonic cleaning. For the different fruit and vegetable cleaning processes, it is necessary to consider the combined effects of the various parameters in order to find the optimum cleaning solution. Future research will require using ultrasonic technology at a commercial level in the food industry, thus optimizing procedures and scaling-up to commercial production. The commercial application of ultrasound to improve the microbial safety of whole fresh products remains a challenge.

## Figures and Tables

**Figure 1 foods-11-02874-f001:**
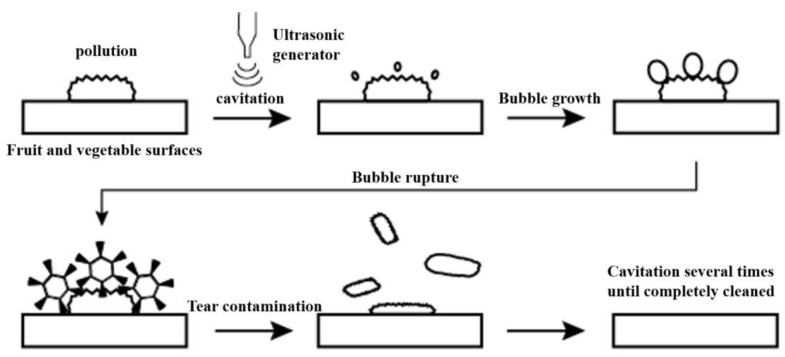
Ultrasonic cleaning process.

**Figure 2 foods-11-02874-f002:**
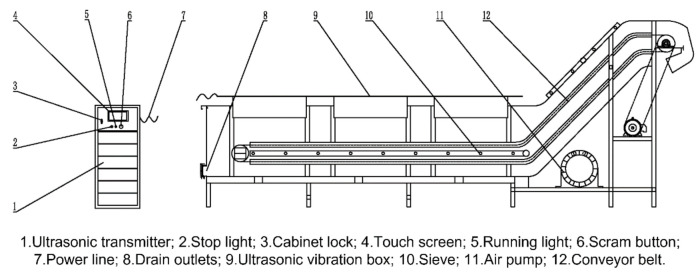
Structural diagram of ultrasonic cleaning machine.

**Figure 3 foods-11-02874-f003:**
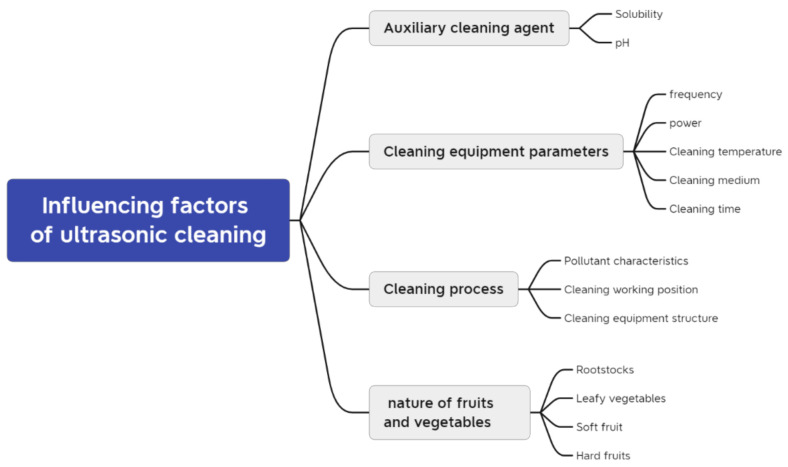
Influencing factors of ultrasonic cleaning.

**Figure 4 foods-11-02874-f004:**
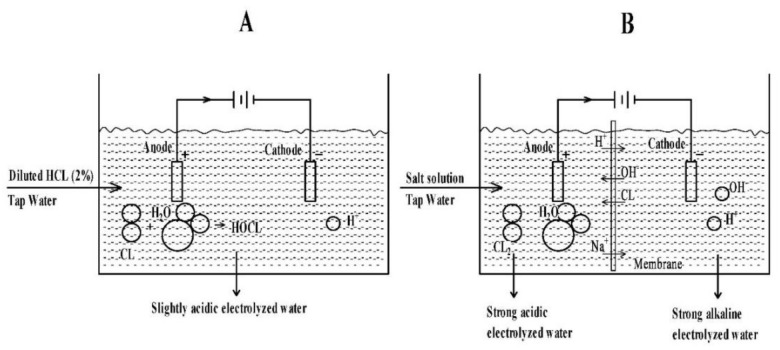
Schematic diagram of electrolyzed water production processes. (**A**) Slightly-acidic electrolyzed water, (**B**) acidic electrolyzed water, and essential electrolyzed water [98].

**Table 1 foods-11-02874-t001:** Ultrasound cleaning technology of whole fruits and vegetables.

Food	Microorganism	Methods	Reduction (log10 CFU/g Sample)	Cleaning Effect(Y/N)	References
**Strawberry**	*E. coli*	32 kHz, 10 W/L, 600 s, surfactant	1.96	Y	[48]
	Total mesophiles	35 kHz, 21.4 W/L, 120 s, 65 °C	8.24	Y	[49]
	TVC, YMC	120 W, 35 kHz, 15 °C Sample/water: 1/25	0.6,1.4	Y	[50]
	TVC, YMC	350 W/L, 40 kHz, 20 °C, 10 min	0.6,0.5	Y	[51]
	*E Coli*, *S. aureus*, *S. Enteritidis* and *L. innocua*	37 kHz, 30 W/L, 3600 s, BPW, 24 kJ/m^2^, 1200 s	3.04, 2.42, 5.52, 6.12, 2.73–3.98	Y	[52,53]
	Total bacteria	33 kHz, 60 W	2	Y	[54]
**Cabbage**	*E. coli*	40 kHz,300 W, 20–30min	>3	Y	[55]
	*E. coli*	32 kHz, 10 W/L, 600 s, surfactant	2.91	Y	[56]
	*L. monocytogenes*	40 kHz, 400 W/L, 180 s, 40 °C	2.8–3.11	Y	[57]
	mesophilic aerobic bacteria	20–60 kHz, 300 W, 10 min	0.7	Y	[58]
**Lettuce**	*S. enterica*	26 kHz, 200 W, 5 min	2.23	Y	[59]
	*E. coli O157:H7*	280 W/L, 20 kHz, 53 min	4.4	Y	[60]
	*E. coli* and *L. monocytogenes*	40 kHz, 400 W/L, 180 s, SAEW	2.5–2.8, 2.6	Y	[61]
	*E. coli*, *S. Typhimurium* and *L. monocytogenes*	40 kHz, 30 W/L, 300 s, organic acids	2.75, 3.18 2.87	Y	[62]
	*E. Coli*, *S. aureus*, *S. Enteritidis* and *L. innocua*	37 kHz, 30 W/L, 3600 s, 1200 s	2.3, 1.7, 5.72 1.88, 1.75–2.85	Y	[52,53]
	*S. enterica* and *E. coli*	26 kHz, 200 W, 90 s, essential oils	1.68–3.08,0.76–2.65	Y	[63,64]
	*E. coli* and *S. Typhimurium*	20 kHz, 131.25 W/L, NNEW	4.4,4.3	Y	[65]
	*S. Typhimurium* and *E. coli*	32 kHz, 10 W/L, 600 s, surfactant	2.7, 2.11	Y	[48]
**Spinach**	*E. coli* and *L. monocytogenes*	40 kHz, 400 W/L, 180 s, SAEW	2.41, 2.49	Y	[61]
	*E. coli O157:H7*	200 W/L, 21.2 kHz, 2 min, acidified sodium chloride (200 mg/L)	4	Y	[66]
	*E. coli*	25 kHz, 79.41 W/L, 60 s	4.45	Y	[67]
**Carrots**	*Bacillus cereus* spores	40 kHz, 0.1% Tween 20, 20 °C, 5 min	2.22	Y	[68]
**Tomato**	*E. coli O157:H7*	20 kHz, 130–210 W, 5–15 min	2.88–4.22	Y	[65]
	*E. coli*, *S. Typhimurium*	20 kHz, 131.25 W/L, NNEW	no detection	Y	[65]
	*S. enterica*, aerobic mesophile	45 kHz, 600 s + peracetic acid	3.90, 4.44	Y	[69]
**Cherry tomatoes**	Aerobic mesophiles and yeasts and molds	20/40 kHz, 300 W,10 min	>2	Y	[70]
	mesophilic aerobic bacteria, and mold and yeast	20–60 kHz, 300W, 25 min	2–4	Y	[71]
	spoilage bacteria	20 kHz, 800 W,4–8 min	0.42–1.04	Y	[72]
	*S. enterica* Typhimurium	45 kHz, 10–30 min	0.83–1.73	Y	[69]
**Japanese plum**	*Aerobic mesophilic bacteria*	US 40 kHz, ClO_2_,10 min, 20 °C	3	Y	[11]
**Peach fruit**	*Penicillium expansum*	US 40 kHz þ salicylic acid, 10 min, 20 °C	Reduce blue mold inpeach fruit	Y	[73]
**Apples**	*E. coli* and *S. enterica*	170 kHz, 20 ppm ClO_2_, 360 s	3.3, 4	Y	[56]
	Psychrophilic and mesophilic bacteria	35 kHz, 72–840W, VI	1.0–2.6	Y	[74]
**Broccoli**	*E. coli O157:H7*	40 kHz,30 min/23 °C	1.04	Y	[75]
**Cauliflower**	*Hemipychomycota fungi*	20–40 kHz, 40 W/L, 15 min.	2	Y	[76]
**Alfalfa**	*E. coli* and *S. enterica*	26 kHz, 200 W, 5 min	1.40 and 1.06	Y	[77]
**Cucumber**	*C. sakazakii*	37 kHz, 13.57 W/L, PA	0.60–3.51	Y	[78]
**Melons**	*E. coli* and *S. enterica* Enteritidis	40 kHz, 120 s, 1% lactic acid	2.5, 3.1	Y	[79]
**Red bell pepper**	*L. innocua*	35 kHz, 120 W, 15 °C	1.98	Y	[80]
**Iceberg lettuce**	*S. Typhimurium*	32–40 kHz, 10 W/L, 10 min	1.5	Y	[48]

**Table 2 foods-11-02874-t002:** Characteristics of disinfectants for ultrasonic cleaning.

Disinfectants	Mechanism	Food	Results	Conclusion	References
Chlorine dioxide	Destroy amino acids of microorganisms	plum	Significant reduction in surface bacteria	Effectively inactivate the bacteria and viruses	[89]
		bean sprouts	Significant reduction in *E. coli* and *Salmonella*		[77]
		apples	Reduced rate of loss of vitamin C and flavonoids		[11]
Ozone	Destroy microbial cell membrane	tomatoes	Effective inactivation of microorganisms	High disinfection efficiency, no secondary contamination and reduced pesticide residues, but use with extra care	[92]
		strawberries	Better quality obtained		[93]
		lettuce	Significantly reduces pesticide residues		[94]
Electrolyzed water	Destroy cell membranes of microorganisms, destroy electron respiratory transport chain of microorganisms	broccoli	Significant reduction in surface bacteria	A new and potentially more environmentally friendly sterilization technology that accelerates bacterial death and reduces early damage to cells	[77]
		cherry tomatoes and strawberries	Significant reduction in surface bacteria		[102]
Non-thermal plasma	Alter chemical properties or structures of biomolecules	apples	Effective reduction of *E. coli* and *Salmonella*	Effective in killing bacteria on its own, but less studied in combination with ultrasound on fruits and vegetables	[108]

## Data Availability

Data is contained within the review.

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
