# Peer review of "Use of Ultrasonic Cleaning Technology in the Whole Process of Fruit and Vegetable Processing"

_foods, 2022, doi:10.3390/foods11182874_

Round 1

Reviewer 1 Report

This paper provides an overview of ultrasonic cleaning technology, ultrasonic cleaning equipment components, fruit, and vegetable cleaning process implementation, residual microbial analysis, ultrasonic-assisted cleaning technology, and the pitfalls of ultrasonic cleaning technology. There are many more detailed works in the literature. However, they were not reported/referenced in the manuscript. I do not believe the manuscript in this form has suitable information for the topic. Therefore, I must reject the manuscript. 

Reviewer 2 Report

The manuscript is written with clear understanding of the project addressed. However, there are major concerns that need to be addressed to enhance the quality of the manuscript. My specific comments are as follows:

Line 8-9: Please rephrase. It is not appropriate to start an abstract with

"Due...". Sentence to be revised.

I recommend adding some commercial examples and regulatory aspects.

In my opinion, authors should also include a section on the sensory aspects and consumer acceptability of ultrasonic-cleaned food products.

Please provide future prospects or trends in ultrasonic cleaning technology for fruits and vegetables.

Please check the format, reference styles, and grammar of the manuscript.

Reviewer 3 Report

The manuscript describes advantages and disadvantages of ultrasonic cleaning technology for fruits and vegetables. However, there are many grammatical and scientific mistakes. The manuscript should be revised by a native for improving English language. Furthermore, it is necessary that the manuscript is precisely revised in some sections. For instance, the effect of ultrasound cleaning on storage of fruits and vegetables has not been discussed.

Some points for improving the manuscript are as follow:

Line 9 what do mean about food related problems?

Line 11: is this result of the previous sentence? Please be conscious about first sentence which is not directly related to Ultrasonic. 

Line 24-24: Please amend it, both grammatical and scientific; development of better ultrasonic cleaning technology is the key technology???

Line 32: all aspects of life??

Line 65: Ultrasound…

Line 94-110 if you are talking about drawbacks, you must explain clearly in all aspects. It should be amended. What are the drawbacks?

Line 126: In first sentence you are talking about probes. In second sentence traditional method…. What is relation. Paragraph should be amended. 

Lines 272-300 : It should also be mentioned that reduction of microorganisms are not exactly because of killing. The majority of reduction is due to detachment of microorganisms and contaminations from surface of fruits. It should be discussed correctly.  

Section 4.1 The influence of various factors on ultrasonic cleaning; The influence of the nature of fruits and vegetables on cleaning process should also be discussed. 

Lines 336-340: Writing should be amended. It should be explained that ultrasonic technology had some disadvantages such as decrease of some characteristics in the food and etc. Line 339 the last sentence should also be written in passive form. Furthermore, the reason that authors believe auxiliary reagent cleaning is useful should also be described, briefly.

Line 346: please write scientific names in italic. It should be accomplished throughout the manuscript.  

Line 351 Why did you write “Hygienic standard” in capital form?

Lines 392-393: what is the meaning of sentence?? 0.1 ppm of what?

There are many mistakes in writing of scientific names. For instance, line 437.

Line 457; to optimize.

Round 2

Reviewer 1 Report

The MS in this form is suitable for publication.

Author Response

We thank the reviewers for your comments and the authors have completed further revisions as requested by the editors.

Reviewer 2 Report

Well done. The manuscript are written with clear understanding of the project addressed. Acceptance is suggested.

Author Response

(The authors gave the same response as above.)

Reviewer 3 Report

please check correct form of scientific names, especially in tables.
